# CHARACTER-AWARE ATTENTION RESIDUAL NETWORK FOR SENTENCE REPRESENTATION

**Xin Zheng**
Nanyang Technological University, Singapore
SAP Innovation Center, Singapore
xzheng008@e.ntu.edu.sg, xin.zheng@sap.com

**Zhenzhou Wu** *
SAP
Singapore
zhenzhou.wu@sap.com

## ABSTRACT

Text classification in general is a well studied area. However, classifying short and noisy text remains challenging. Feature sparsity is a major issue. The quality of document representation here has a great impact on the classification accuracy. Existing methods represent text using bag-of-word model, with TFIDF or other weighting schemes. Recently word embedding and even document embedding are proposed to represent text. The purpose is to capture features at both word level and sentence level. However, the character level information are usually ignored. In this paper, we take word morphology and word semantic meaning into consideration, which are represented by character-aware embedding and word distributed embedding. By concatenating both character-level and word distributed embedding together and arranging words in order, a sentence representation matrix could be obtained. To overcome data sparsity problem of short text, sentence representation vector is then derived based on different views from sentence representation matrix. The various views contributes to the construction of an enriched sentence embedding. We employ a residual network on the sentence embedding to get a consistent and refined sentence representation. Evaluated on a few short text datasets, our model outperforms state-of-the-art models.

## 1 INTRODUCTION

For text classification, a popular feature representation method is bag-of-word. However, this representation has an intrinsic disadvantage that two separate features will be generated for two words with the same root or of different tenses. Lemmatization and stemming could be applied to partially address this problem, but may not always leads to correct results. For example, "meaningful" and "meaningless" would both be considered as "meaning" after applying lemmatization or stemming algorithms, while they are of opposite meanings. Thus, word morphology could also provide useful information in document understanding, particular in short text where the information redundancy is low.

For short text, an important issue is data sparsity, particularly when utilizing feature representation method like bag-of-word, regardless the weighting scheme. Therefore, various distributed word representation like Word2Vec (Mikolov et al., 2013) and document representation Doc2Vec (Le & Mikolov, 2014) have been proposed to address the problem. However, this kind of method miss the word morphology information and word combination information. To deal with these issues, we propose a model which could capture various kinds of features that could benefit classification task.

In this paper, we look deep into characters. We learn character representation and combine both character-level (Zhang et al., 2015) and word-level embedding to represent a word. Thus both morphology and semantic properties of the word are captured. As we know, not all the words in a sentence contribute the same when predicting the sentence's label. Therefore, highlight the relatively pertinent information would give better chance of correct prediction. Attention mechanism (Mnih et al., 2014; Bahdanau et al., 2016) which focuses on specific part of input could help achieve this goal. The applications of attention mechanism are mostly on sequential model, while we employ

---

*The two authors contribute the same for the work.

the idea of attention on a feed-forward network (Raffel & Ellis, 2015). By multiplying the weight assigned by attention mechanism to its corresponding word vector, a weighted feature matrix could be constructed by concatenating the sequence of word embeddings in a sentence.

Short text usually could not provide much useful information for class prediction. We try different views to extract as much information as possible to construct an enriched sentence representation vector. Specifically, to convert a sentence representation matrix to an enriched vector, we draw two types of features. The first one is based on word feature space and the other one is based on n-gram. However, not all the features contribute the same on sentence classification. Attention mechanism is applied to focus on the significant features. Since these features come from different views, we need a method to make the elements consistent. The residual network proposed in (He et al., 2015; 2016) achieve much better results on image classification task. In other words, the residual mechanism could construct better image representation. Therefore, we adopt residual network to refine the sentence representation vector. Once we obtain a good quality representation for the sentence, it will be delivered to a classifier.

## 2 RELATED WORK

There are many traditional machine learning methods for text classification and most of them could achieve quite good results on formal text datasets. Recently, many deep learning methods are proposed to solve the text classification task (Zhang et al., 2015; dos Santos & Gatti, 2014; Kim, 2014).

Deep convolutional neural network suggests benefits in image classification (Krizhevsky et al., 2012; Sermanet et al., 2013). Therefore, many research also try to apply it on text classification problem. Kim (2014) propose a model similar to Collobert et al. (2011) architecture. However, they employ two channels of word vectors. One is static throughout training and the other is fine-tuned via back-propagation. Various size of filters are conducted on both channel, and the results are concatenated together. Then max-pooling over time is taken to select the most significant feature among each filter. The selected features are concatenated as the sentence vector.

Similarly, Zhang et al. (2015) also employ the convolutional networks but add character-level information for text classification. They design two networks, one large and one small. Both of them have nine layers including six convolutional layer and three fully-connected layers. Between the three fully connected layers they insert two dropout for regularization. For both convolution and max-pooling layers, they employ 1-D version (Boureau et al., 2010). After convolution, they add the sum over all the results from one filter as the output. Specially, they claim 1-D max-pooling enable them to train a relatively deep network (Boureau et al.).

Besides applying models directly on testing datasets, more aspects are considered when extracting features. Character-level feature is adopted in many tasks besides Zhang et al. (2015) and most of them achieve quite good performance. dos Santos & Zadrozny (2014) take word morphology and shape into consideration which have been ignored for part-of-speech tagging task. They suggest the intra-word information is extremely useful when dealing with morphologically rich languages. They adopt neural network model to learn the character-level representation which is further delivered to help word embedding learning. Kim et al. (2016) construct neural language model by analysis of word representation obtained from character composition. Results suggest the model could be encode semantic and orthographic information from character level.

Attention model is also utilized in our model, which is used to assign weights for each parts of components. Usually, attention model is used in sequential model (Rocktäschel et al., 2015; Mnih et al., 2014; Bahdanau et al., 2016; Kadlec et al., 2016). The attention mechanism includes sensor, internal state, actions, and reward. At each time, the sensor will capture a glimpse network which only focus on a small part of the network. Internal state will summarize the extracted information. Actions decides the location for the next step and reward suggests the benefit when taking the action. In our condition, we adopt a simplified attention network as (Raffel & Ellis, 2015; 2016). We do not need to guess the next step location and just give a weight on each components which indicates the significance of the element.

Residual network (He et al., 2015; 2016; Chen et al., 2016) is known to be able to make neural network deeper and relieve degradation problem at the same time. And residual network in (He et al., 2015) outperforms the state-of-the-art models on image recognition. He et al. (2016) introduces

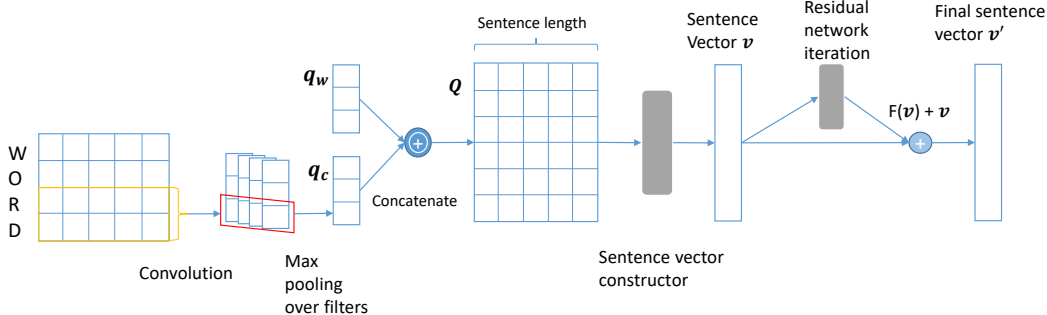

Figure 1: Illustration of the proposed model. $q_c$ is the character-level embedding vector for the word, and $q_w$ is the word embedding generated according to (Mikolov et al., 2013). Column of $\mathbf{Q}$ is the concatenation of $q_c$ and $q_w$. The row length is sentence length. The grey box for sentence vector constructor is illustrated in Figure 2 and the grey box for Residual network iteration is illustrated in Figure 3(b).

how to make the residual block more efficient on image classification. Similarly, for short text classification problem, the quality of sentence representation is also quite important for the final result. Thus, we try to adopt the residual block as in (He et al., 2015; 2016) to refine the sentence vector.

## 3 CHARACTER-AWARE ATTENTION RESIDUAL NETWORK

In this paper, we propose a character-aware attention residual network to generate sentence representation. Figure 1 illustrates the model. For each word, the word representation vector is constructed by concatenating both character-level embedding and word semantic embedding. Thus a sentence is represented by a matrix. Then two types of features are extracted from the sentence matrix to construct the enriched sentence representation vector for short text. However, not all the features contribute the same for classification. Attention mechanism is employed to target on pertinent parts. To make features extracted from different views consistent, a residual network is adopt to refine the sentence representation vector. Thus, an enriched sentence vector is obtained to do text classification.

### 3.1 WORD REPRESENTATION CONSTRUCTION

Let $\mathcal{C}$ be the vocabulary of characters, and $\mathbf{E} \in \mathbb{R}^{d_c \times |\mathcal{C}|}$ is the character embedding matrix, where $d_c$ is the dimensionality of character embedding. Given a word, which is composed of a sequence of characters $[c_1, c_2, ..., c_{n_c}]$, its corresponding character-level embedding matrix would be $\mathbf{E}^w \in \mathbb{R}^{d_c \times n_c}$. Herein,

$$\mathbf{E}^w_{\cdot i} = \mathbf{E} \cdot v_i \tag{1}$$

where $v_i$ is a binary column vector with 1 only at the $c_i$-th place and 0 for other positions. Here, we fix the word length $d_c$ and take zero-padding when necessary.

For each of such matrix $\mathbf{E}^w$, a convolution operation (Le Cun et al., 1990) with $m$ filters (*i.e.*, kernels) $\mathbf{P} \in \mathbb{R}^{d_c \times k}$ is applied on $\mathbf{E}^w$, and a set of feature maps could be obtained. Instead of adopting max-pooling over time (Collobert et al., 2011), we adopt max-pooling over filters operation to capture local information of words as shown in Figure 1. Similar operation is adopted in (Shen et al., 2014). That is we get the max feature value over results of $m$ filters at the same window position, which depicts the most significant feature over the $k$ characters. Thus, a vector $\mathbf{q^c}$ for the word which captures the character-level information is constructed.

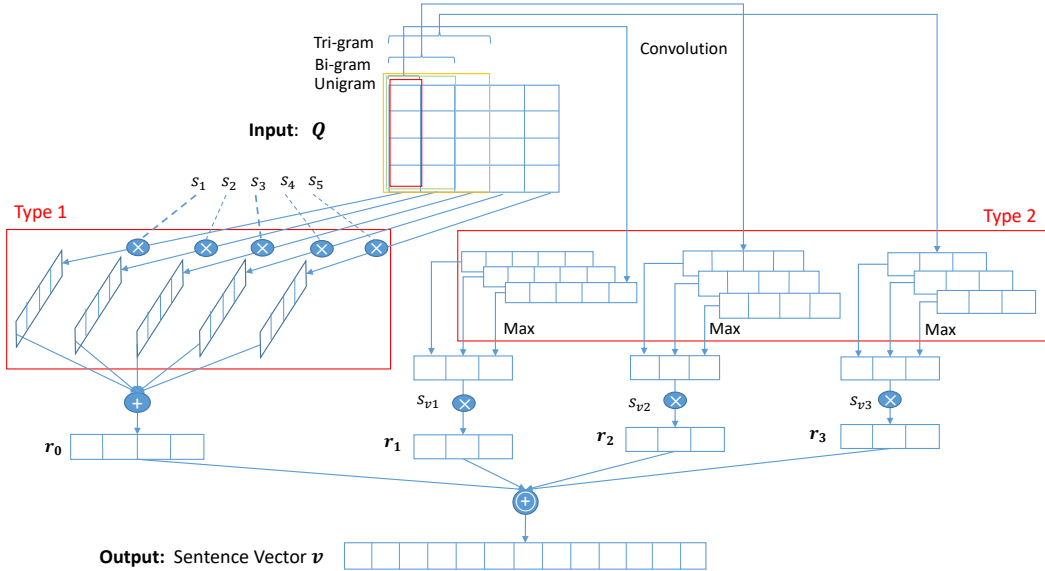

Figure 2: Illustration of sentence representation. The input $\mathbf{Q}$ is from Figure 1. The weights $s_1, s_2, s_3, s_4$ and $s_{v1}, s_{v2}, s_{v3}$ are generated from attention mechanism, which is illustrated in Figure 3(a). Type 1 feature and type 2 feature are detailed in 3.2.

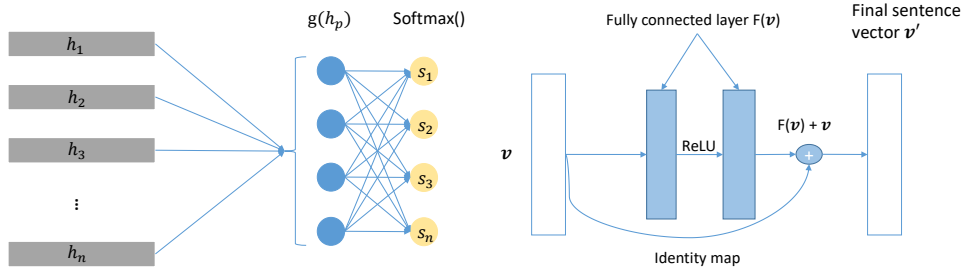

(a) Attention mechanism. This is the basic attention mechanism we used to assign weights $s_1, s_2, s_3, s_4$ and $s_{v1}, s_{v2}, s_{v3}$. The $h_i$ is the corresponding input vector.

(b) Residual block for refining sentence representation $\mathbf{v}$

Figure 3: Illustrations of detailed attention mechanism and residual network.

Note that embedding vector $\mathbf{q^c}$ could only capture the word morphological features, while it can not reflect word semantic and syntactic characteristics. Therefore, we concatenate the distributed word representative vector $\mathbf{q^w}$ (*i.e.,* Word2Vec) (Mikolov et al., 2013) to $\mathbf{q^c}$ as the word's final representation $\mathbf{q} \in \mathbb{R}^{(d_c+d_w)}$, where $d_w$ is the dimensionality of Word2Vec. Given a sentence, which consists of a sequence of words $[w_1, w_2, ..., w_{n_w}]$, its representation matrix is $\mathbf{Q} \in \mathbb{R}^{(d_c+d_w) \times n_w}$.

## 3.2 SENTENCE REPRESENTATION VECTOR CONSTRUCTION

To overcome the lack of information issue for short text, we explore various kinds of useful information from limited context. From higher level, we adopt two types of features as shown in

Figure 2 (*i.e.,* type 1 feature and type 2 feature). They capture different views of information for the short text, which could be considered as results from horizontal view and vertical view on sentence representation matrix $\mathbf{Q}$ separately.

**Type** 1 feature takes word's feature space (*i.e.,* horizontal view on $\mathbf{Q}$) into consideration. The feature space is the composition of both character-level embedding and word semantic embedding. Each word is a point in the feature space. We formulate the summation over all words appearing in the sentence as the sentence's representation, inspired by (Zhang et al., 2015). In fact, not all the words in a sentence contribute the same for prediction. Therefore, we want to highlight the significant words and this is realized by weighting the word's representation features. To assign the weights, we employ attention mechanism, and multiply the weight to the word feature vector as Equation 2. Specifically, we follow Raffel & Ellis (2015) and Bahdanau et al. (2014) as shown in Figure 3(a). For each word representation vector $\mathbf{q_i}$, we apply a Tanh function on the linear transformation of $\mathbf{q_i}$ as $g(\mathbf{q_i}) = Tanh(W_{qh}\mathbf{q_i} + b_{qh})$, where $W_{th} \in \mathcal{R}^{1\times(d_c+d_w)}, b_{th} \in \mathcal{R}$. Then a softmax function on $g(\mathbf{q_i})$ is used to assign a weight $s_i$ for each $\mathbf{q_i}$, which indicates the significance of word $i$ in the sentence.

$$s_i = \frac{\exp(g(\mathbf{q_i}))}{\sum_{n_w}^{j=1} \exp(g(\mathbf{q_j}))}, \tilde{\mathbf{q_i}} = s_i\mathbf{q_i}. \tag{2}$$

As a result, we can get a weighted sentence representation matrix $\tilde{\mathbf{Q}} \in \mathbb{R}^{(d_c+d_w)\times n_w}$. Then we employ an average over words in the sentence at the same feature position and obtain a sentence representation vector $\mathbf{r_0}$.

**Type** 2 feature models the word level features (*i.e.,* vertical view on $\mathbf{Q}$). As we know, sometimes continuous words combination is meaningful and pertinent for sentence classification. To capture n-gram information, we apply convolution operation on $\mathbf{Q}$, which is followed by a max-pooling over time. We adopt several different kernel sizes to model various n-grams. Different n-grams contribute differently. The attention mechanism is utilized again on the vectors of n-gram representations, and the resulting weights indicate their significance. We get the weighted feature vectors $\mathbf{r_1}, \mathbf{r_2}, \mathbf{r_3}$. Concatenating $\mathbf{r_0}, \mathbf{r_1}, \mathbf{r_2}, \mathbf{r_3}$, the complete sentence vector $\mathbf{v}$ is constructed.

### 3.3 RESIDUAL NETWORK FOR REFINING SENTENCE REPRESENTATION

The residual learning (He et al., 2015; 2016) is reported to outperform state-of-the-art models in image classification task and object detection task. This suggests residual learning could help to capture and refine the embedding. To make the features of sentence vector $\mathbf{v}$ from different views consistent, we employ residual learning to $\mathbf{v}$.

Let the desired mapping as $\mathcal{H}(\mathbf{v})$, instead of making each layer directly optimize $\mathcal{H}(\mathbf{v})$, residual learning (He et al., 2015) turns to fit the residual function:

$$\mathcal{F}(\mathbf{v}) := \mathcal{H}(\mathbf{v}) - \mathbf{v}. \tag{3}$$

Thus, the original target mapping becomes:

$$y = \mathcal{F}(\mathbf{v}) + \mathbf{v}. \tag{4}$$

Both residual function $\mathcal{F}(\mathbf{v})$ and the added input form $\mathbf{v}$ are flexible. In our model, we construct the building block by two fully connected layers connected by a ReLU (Nair & Hinton, 2010) operation as shown in Figure 3(b). Meanwhile, the identity mapping is adopted by performing a shortcut connection and element-wise addition:

$$\mathbf{v}\prime = \mathcal{F}(\mathbf{v}, G) + \mathbf{v} \tag{5}$$

where $\mathbf{v}\prime$ is the refined sentence vector, $G$ is the weight matrix to be learned.

After getting the sentence embedding $\mathbf{v}\prime$ from the building block, it is further delivered to a softmax classifier for text classification.

Table 1: Statistics of datasets.

| Dataset | Classes | Train Samples | Test Samples | Average length of text |
|---------|---------|---------------|--------------|------------------------|
| Tweet | 5 | 28,000 | 7,500 | 7 |
| Question | 5 | 2,000 | 700 | 25 |
| AG_news | 5 | 120,000 | 7,600 | 20 |

## 4 EXPERIMENT

### 4.1 DATASETS

We adopt testing datasets from different sources. There are three datasets, including Tweets, Question, AG_news. All of them are relatively short.

**Tweets** are typical short text with only 140 characters limitation. We crawl the tweets from Twitter with a set of keywords, which is specifically about some products. We label them as positive, negative, neutral, question and spam.

**Question** dataset is a small dataset. The content is short questions, and the labels are question types.

**AG_News** dataset is from (Zhang et al., 2015). The reason we choose this is because the length of text is much shorter than others. The news here only contains the title and description fields.

### 4.2 EXPERIMENT SETTINGS

In this paper, we take 128 ASCII characters as character set, by which most of the testing documents are composite. We define word length $n_c$ as 20 and character embedding length $d_c$ as 100. If a word with characters less than 20, zero padding is taken. If the length is larger than 20, just take the first 20 characters. We train the word distributed embedding using training data and the feature dimension is 300. We take sentence length as 20, which is enough to cover most of crucial words. We add 5 residual blocks to refine the sentence vector.

Table 2: Kernel size for convolutional layers

| Convolutional layer | Kernel size |
|---------------------|-------------|
| Conv:character embedding | $(d_c, 4)$ |
| Conv:ngram | $(d_c + d_w, 1), (d_c + d_w, 2), (d_c + d_w, 3)$ |

### 4.3 BASELINE MODEL

We select both traditional models and deep learning models on classification as baselines.

**TF-SVM** is the bag-of-word feature weighted by counting the term frequency in a sentence. Then deliver the feature matrix to a SVM classifier.

**TFIDF-SVM** is taken as traditional baseline model. Since SVM classifier is robust and state-of-the-art traditional classifier, and TFIDF usually assign good weights for bag-of-words in documents even for tough inputs. So this is a competitive baseline model.

**Lg. Conv, Sm. Conv**   are proposed in (Zhang et al., 2015) which also consider character-level embedding, and they concatenate all the characters' embeddings in a sentence in order as sentence's representation matrix. For fair comparison, we do not include thesaurus to help clear documents.

## 4.4   COMPARISON RESULTS

Table 3 shows the comparison results on the testing datasets. As we can see, the proposed model could outperform baseline models on Tweets and Question datasets. For AG_news dataset, our method could give comparable results as the best baseline model, TFIDF-SVM. The TFIDF-SVM model can achieve relatively better results than others. However, both Lg. Conv and Sm. Conv do not perform well on Tweets dataset. This may because these two models are relatively deep network with several down sampling operations (*i.e.,* max-pooling) and this dramatically decreases the short text representation. And short text does not contain much information. Thus Lg. Conv and Sm. Conv could not give good results. The TF-SVM model also does not perform well on Tweets dataset. This may because the tweet text is too short and term frequencies are mostly 1 which is not enough to provide information on classification. Similar to result of CAR-1 on Tweets data. When removing type 1 feature, the performance drops dramatically. However, for other datasets, in which the document length is longer and the content is relatively formal, removing type 1 feature does not influence the performance that much. Hence, these results suggest the word character-level feature and semantic feature (*i.e.,* type 1 feature) are rather important for short, free-style text. On the other hand, by adding type 2 features can also improve the performance according to results of CAR-2. Consequently, when dealing with short text, either formal or informal, including character-level feature, word-semantic feature and n-gram feature would benefit the performance.

Another comparison is adding the residual network or not. As we can see from Table 3, residual network could refine the vector representation. When removing residual block, performances on three datasets all decrease. In particular, the improvement for shorter and noisy text (Tweets dataset) is more than those relatively longer and formatted documents. Thus, for short noisy text classification problem, one adopts residual building block would improve the performance.

Table 3: Comparison results on accuracy. "CAR" is the proposed model which indicates Character-aware Attention Residual network. "WAR" is the proposed model with only word embedding from Word2Vec. "CA" is "CAR" removing residual network. "CAR-1" is "CAR" removing type 1 feature. "CAR-2" is "CAR" removing type 2 feature. "CAR-1w" is "CAR" removing attention weight assigned to type 1 feature.
.

| Method | Tweets | Question | AG_news |
|---|---|---|---|
| BoW-SVM | 40.34 | 86.35 | 88.77 |
| TFIDF-SVM | 79.96 | 88.85 | **90.89** |
| Lg. Conv | 24.13 | 83.55 | 87.18 |
| Sm. Conv | 40.51 | 88.57 | 84.35 |
| WAR | 72.50 | 89.97 | 89.18 |
| CA | 79.44 | 90.25 | 89.11 |
| CAR-1 | 40.32 | 88.57 | 25.12 |
| CAR-2 | 75.11 | 88.71 | 89.37 |
| CAR-1w | 78.74 | 90.39 | 90.19 |
| CAR | **81.26** | **90.95** | 90.45 |

## 5   CONCLUSION

We propose a character-aware attention residual network for short text classification. We construct the sentence representation vector by two kinds of features. The first is focusing on feature space, which include both character-level characteristics and semantic characteristics. The other is n-gram features. To make them consistent, the residual network helps refine the vector representation. Experiment results suggest both extracted features and the residual network helps on short text clas-

sification. Our proposed method could outperform the state-of-the-art traditional models and deep learning models.

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
