# Peer review of "Character-aware Attention Residual Network for Sentence Representation"

_ICLR 2017 — rejected_

[Public Comment · Tara N Sainath · 15 Nov 2016]
**duplicate paper**

Hi Authors,

You seem to have submitted two of the same paper? Pls advise which is the correct one

Character-aware Attention Residual Network for Sentence Representation
Xin Zheng, Zhenzhou Wu
5 Nov 2016

CHARACTER-AWARE RESIDUAL NETWORK FOR SENTENCE REPRESENTATION
Xin Zheng, Zhenzhou Wu
4 Nov 2016

[Official Review · AnonReviewer1 · rating 4 · confidence 5 · 16 Dec 2016]
**A paper that needs more work**

This paper proposes a character-aware attention residual network for sentence embedding. Several text classification tasks are used to evaluate the effectiveness of the proposed model. On two of the three tasks, the residual network outforms a few baselines, but couldn't beat the simple TFIDF-SVM on the last one.

This work is not novel enough. Character information has been applied in many previously published work, as cited by the authors. Residual network is also not new.

Why not testing the model on a few more widely used datasets for short text classification, such as TREC? More competitive baselines can be compared to. Also, it's not clear how the "Question" dataset was created and which domain it is.

Last, it is surprising that the format of citations throughout the paper is all wrong. 

For example:
like Word2Vec Mikolov et al. (2013)
->
like Word2Vec (Mikolov et al., 2013)

The citations can't just mix with the normal text. Please refer to other published papers.

[Official Review · AnonReviewer3 · rating 4 · confidence 4 · 19 Dec 2016]

This paper proposes a new model for sentence classification. 

Pros:
- Some interesting architecture choices in the network.

Cons:
- No evaluation of the architecture choices. An ablation study is critical here to understand what is important and what is not.
- No evaluation on standard datasets. On the only pre-existing dataset evaluated on a simple TFIDF-SVM method is state-of-the-art, so results are unconvincing.

[Official Review · AnonReviewer2 · rating 4 · confidence 4 · 19 Dec 2016]
**Need more explanation about network architecture**

This paper proposes a new neural network model for sentence representation. This new model is inspired by the success of residual network in Computer Vision and some observation of word morphology in Natural Language Processing. Although this paper shows that this new model could give the best results on several datasets, it lacks a strong evidence/intuition/motivation to support the network architecture.

To be specific:

- I was confused by the contribution of this paper: character-aware word embedding or residual network or both?
- The claim of using residual network in section 3.3 seems pretty thin, since it ignores some fundamental difference between image representation and sentence representation. Even though the results show that adding residual network could help, I was still not be convinced. Is there any explanation about what is captured in the residual component from the perspective of sentence modeling?
- This paper combines several components in the classification framework, including character-aware model for word embedding, residual network and attention weight in Type 1 feature. I would like to see the contribution from each of them to the final performance, while in Table 3 I only saw one of them. Is it possible to add more results on the ablation test?
- In equation (5), what is the meaning of $i$ in $G_i$?
- The citation format is impropriate

[Final Decision · Program Chairs · 06 Feb 2017]
**ICLR committee final decision**

The paper introduces some interesting architectural ideas for character-aware sequence modelling. However, as pointed out by reviewers and from my own reading of the paper, this paper fails badly on the evaluation front. First, some of the evaluation tasks are poorly defined (e.g. question task). Second, the tasks look fairly simple, whereas there are "standard" tasks such as language modelling datasets (one of the reviewers suggests TREC, but other datasets such as NANT, PTB, or even the Billion Word Corpus) which could be used here. Finally, the benchmarks presented against are weak. There are several character-aware language models which obtain robust results on LM data which could readily be adapted to sentence representation learning, eg. Ling et al. 2016, or Chung et al. 2016, which should have been compared against. The authors should look at the evaluations in these papers and consider them for a future version of this paper. As it stands, I cannot recommend acceptance in its current form.